# Women’s Health: Contemporary Management of MS in Pregnancy and Post-Partum

**DOI:** 10.3390/biomedicines7020032

**Published:** 2019-04-19

**Authors:** Kelly Tisovic, Lilyana Amezcua

**Affiliations:** Department of Neurology, Keck School of Medicine, University of Southern California, Los Angeles, CA 90033, USA; tisovic@usc.edu

**Keywords:** multiple sclerosis, pregnancy, post-partum, treatment

## Abstract

Multiple sclerosis (MS) primarily affects women in childbearing age and is associated with an increased risk of adverse post-partum outcomes. Relapses and now fetal exposure to disease modifying treatments in the early phase of pregnancy and thereafter are of concern. Safe and effective contraception is required for women who wish to delay or avoid pregnancy while on disease-modifying treatments. Counseling and planning is essential to assess the risk of both fetal and maternal complications, particularly now in the era of highly efficient and riskier therapies. The purpose of this review is to provide a practical framework using the available data surrounding pregnancy in MS with the goal of optimizing outcomes during this phase in MS.

## 1. Introduction

Multiple sclerosis (MS) is an autoimmune disease of the central nervous system (CNS) characterized by inflammation and demyelination with almost 1 million affected in the US [1]. The disease predominantly affects females and often starts in childbearing age [1], making this period a particularly important stage for treatment decisions. Evidence supporting a reduction of disease activity during pregnancy [2] is likely contributing to the recent observation that more women with MS are considering pregnancy [3]. Because most disease-modifying treatments (DMT) are contraindicated during pregnancy, understanding the risk behind discontinuation and early pregnancy exposure is important. In the current era, a neurologist can now be faced with concerns relating to pregnancy, including contraception, fertility, relapses during pregnancy and post-partum, and breastfeeding.

The purpose of this paper is to provide a practical clinical framework to assist physicians in the treatment of women with MS surrounding pregnancy by reviewing the most recent data surrounding pre-conception care, pregnancy and the post-partum phase in MS with the goal of facilitating informed decisions.

## 2. Methods

We performed a literature search in Pubmed using the search term “multiple sclerosis,” with additional search terms of “pregnancy,” “breastfeeding,” “contraception,” “fertility,” “disease modifying therapy,” “pregnancy outcomes,” “guidelines,” and “counseling” in the last 10 years. The studies felt to be most relevant for clinical practice were selected for inclusion in this review.

## 3. Pre-Pregnancy

### 3.1. Family Planning and Counseling

The rates of pregnancies in MS patients has increased in the US from 2006 to 2014, from 7.91% to 9.47%, while the rate of women without MS decreased from 8.83% to 7.75% over this time [3]. Because there is a high rate of unintentional pregnancies [4] and safety concerns surrounding certain MS DMTs in pregnancy [5], counseling should be performed for all women with MS of reproductive age. In addition, the neurologist should be aware of other related issues such as contraception, fertility, and the low risk of transmission of MS to offspring (2–3% if one parent is affected and about 20% if both parents are affected) [5]. Once a patient has decided to become pregnant, a number of steps can be taken in an effort to achieve optimal outcomes. These are described below, and are outlined in Figure 1.

#### 3.1.1. MS Disease Activity Assessment 

The current MS disease activity should be evaluated. Appropriate laboratory analysis and MRIs should be included in this assessment. For those with active MS disease activity, postponing attempted conception until disease is stable for at least six months is recommended. 

#### 3.1.2. Medication Reconciliation 

The current symptomatic MS medications a patient is taking and their safety in pregnancy should be evaluated. This should be performed with the assistance of a patient’s obstetrician-gynecologist whenever possible. Current DMT use should be discussed in regards to safety and optimal discontinuation timing, which is typically 5 maximal half lives of the DMT, but this time frame may vary in specific circumstances [6]. Standard prenatal medications including prenatal vitamins and folic acid supplementation is advised [6]. Optimal Vitamin D supplementation is important to maintain throughout pregnancy, as an increased risk of MS was seen in offspring of women with low gestational Vitamin D (25(OH)D) levels of less than 30nmol/L in a Finnish study [7]. Smoking cessation is recommended for all patients, given its impact on both MS disease activity and on pregnancy [6].

#### 3.1.3. Anticipation of Issues Encountered during Pregnancy and Post-partum

Physicians should discuss the potential for MS symptoms to worsen or for new MS symptoms to appear, especially during the less-protective phases of pregnancy and especially in patients with highly active disease or on DMTs with risk for rebound activity. Plans regarding breastfeeding should also be discussed pre-conception, as this will affect the timing of DMT resumption. 

### 3.2. Oral Contraceptives and Multiple Sclerosis (MS) Disease Activity

Estrogen has known anti-inflammatory properties and has shown to be neuroprotective in preclinical studies [8]. The reduction in disease activity observed during pregnancy is thought to be related to high levels of estriol, especially in the third trimester [9]. In experimental autoimmune encephalomyelitis (EAE) models, administration of estriol improved EAE, which correlated with a decrease in the number of CNS inflammatory cells [8]. However, retrospective and prospective studies have reported mixed results regarding oral contraceptive (OC) use and MS risk [10,11,12] and have reported a positive influence on relapse rates [13]. While Hellwig et al. found that OC use was associated with a slightly increased risk of MS/clinically isolated syndrome (CIS) (adjusted odds ratio (OR) = 1.52, 95% CI = 1.21–1.91; *p* < 0.001) [12], the risk did not change with duration of OC use, suggesting non-causal association. Other studies have supported a positive effect. Rejali et al. found a statistically significant relationship between history of OC use (OR = 0.492, *p* = 0.002) and MS risk and in the duration of OC use (OR = 0.881, *p* = 0.008) and MS risk [11]. Holmqvist et al. found the mean age of MS onset was significantly higher in patients with OC use prior to MS onset than those without OC use (26 years old vs. 19 years old, *p* < 0.001). Additionally, age of MS onset increased with increasing time of OC use prior to MS onset [10]. Bove et al. evaluated effects of past-, current- or never-OC use in women with new onset relapsing remitting multiple sclerosis (RRMS) or CIS started on a first-line injectable disease-modifying therapy and found that past OC users had a statistically significant lower annualized relapse rate (ARR) compared to never OC users (Relative Risk (RR) = 0.64, *p* = 0.031) and that current OC users had a non-statistically significant lower ARR compared to never OC users (RR = 0.97, *p* = 0.91) [13]. 

The anti-inflammatory effects of estradiol in combination with injectable MS therapies have been demonstrated in preliminary studies [9,14]. Pozzilli et al. combined low-dose and high-dose oral contraceptives with interferon beta and evaluated disease activity via cumulative number of combined unique active (CUA) lesions on magnetic resonance imagining (MRI) at 96 weeks. There was a 26.5% (*p* = 0.04) reduction in CUA lesions in the high dose OC group compared to interferon beta alone and a non-significant reduction in the low dose OC group compared to interferon beta alone [14]. Voskuhl et al. published a randomized, double-blind, placebo controlled phase 2 trial to assess the safety and efficacy of estriol and progesterone in combination to glatiramer acetate versus glatiramer acetate alone using a primary endpoint of annualized confirmed relapse rate at 24 months, and using a significance level of *p* = 0.01. Confirmed relapse rate was 0.25 (95% CI 0.17–0.37) relapses per year in the estriol treated group compared to 0.37 (95% CI 0.25–0.53) relapses per year in the placebo group with an adjusted rate ratio of 0.63 (95% CI 0.37–1.05, *p* = 0.077) [9]. Larger studies with longer treatment times are needed to better evaluate the effect of OC treatment and to help better stratify the risk-benefit of long term estrogen use. 

Clinical considerations: reliable contraception is recommended for patients taking DMT. DMT-specific recommendations for contraceptive use are included in Table 1 [15,16,17,18,19,20], if provided specifically in the prescribing recommendations. Beyond the need for pregnancy protection while using DMT, oral contraceptive use is considered safe in MS [13], and potential additional benefits on disease activity may be seen with oral contraceptive use in combination with platform injectable therapy [9,14].

### 3.3. Fertility

#### 3.3.1. Effects of MS on Fertility

There are multiple observational studies supporting the argument that fertility may be influenced by MS [21,22,23,24,25]. An Italian study found that women with MS are more frequently childless compared to the general population, with reported rates of 22% vs. 13%, respectively [21]. However, it is unclear if childlessness is due to behavioral factors such as disability, fear or beliefs about caring for a newborn child or transmitting MS to her offspring, or actually a disease-related pathology [26]. Another important factor impacting fertility in MS patients includes sexual dysfunction, which is very common, reported in 30–70% of MS patients [26] and is beyond the scope of this paper. 

Reassuringly, an observational study in a French cohort evaluated fecundity, defined as the time to become pregnant, in MS patients both before and after MS onset [22]. No differences in the time to conception prior to or after MS disease onset were found (on average <1 year). However, despite their normal fecundity, the mean number of children per women with MS was less than in the general French population (1.37 versus 1.99, respectively) [22]. It is important to note that infertility in Western populations is not uncommon, estimated at 10–20%, and because MS occurs at a fertile age, it is possible that infertility is not actually MS mediated [27]. Nevertheless, assisted reproductive techniques (ART) have been reported to be more common in women with MS [23].

Other studies have focused on evaluating hormonal differences, where higher levels of prolactin, luteinizing hormone (LH), and follicle-stimulating hormone (FSH) and lower levels of estrogen have been reported among MS women [24]. Elevated FSH in the early follicular phase is an indicator of low ovarian reserve [25]. Another marker of low ovarian reserve, Anti-Mullerian hormone (AMH), has been found to be decreased in reproductive-aged women with other autoimmune diseases and does not fluctuate with menstrual cycles as FSH and LH levels do. To better understand ovarian reserve in women with MS, serum AMH levels were examined in 76 RRMS patients and in 58 healthy controls and found serum AMH levels to be significantly lower in patients with RRMS than in healthy controls, and a higher proportion of RRMS patients showed very low AMH levels [25]. However, in a follow up study, no significant differences in AMH levels were found [28] indicating that larger studies are needed to better understand if lower ovarian reserve is found in reproductive-aged women with MS. Thus, the ability to conceive may be influenced by various factors which makes fertility a concern for many women affected with MS.

If a woman with MS does suffer from infertility, ART should be considered with caution. Gonadotropin-releasing hormone (GnRH) agonists and antagonists are used to suppress the influence of the hypothalamic-pituitary gland axis, preventing an LH surge and thus spontaneous ovulation. Subsequent ovarian hyperstimulation is accomplished by administration of gonadotropins. Following stimulation, controlled ovulation is accomplished by human choriongonadotropin (hCG), and administration of progesterone to support the luteal phase. Fertilization is completed by intrauterine insemination, in vitro fertilization, or intracytoplasmatic sperm injection [27]. 

There are no randomized control trials to evaluate the safety or changes in relapse rate in MS patients who are treated with ART. Despite the clinical improvements seen with the use of GnRH agonists in EAE models [29], their use in humans with MS have shown opposite effects [27]. ART therapy in MS patients may cause increased disease activity, and patients who plan to pursue these therapies should be counseled about this risk sufficiently. Hellwig et al. published a review of five small studies evaluating ART use in MS patients using various reproductive techniques. This collection of studies demonstrated an increase in relapse rate following unsuccessful ART and increased MRI activity. Downregulation of pituitary GnRH receptors via GnRH agonists might account for this increased relapse rate. Additional theories behind worsening MS activity following ART include discontinuation of DMT, stress related to infertility, rapidly changing hormone levels inducing pro-inflammatory changes, and ART-mediated increases in immune cell movement across the blood-brain-barrier via induction of interleukin-8 (IL-8), vascular endothelial growth factor (VEGF), and CXC chemokine ligand 12 (CXCL-12) [27]. 

#### 3.3.2. Effects of Disease-Modifying Treatments (DMTs) on Fertility

In evaluating the potential influence of DMT use on fertility, information on humans is currently scarce. Table 2 summarizes reported effects of DMTs on fertility with special concerns to the following DMTs: interferon beta [30], natalizumab [31], alemtuzumab [17], mitoxantrone [31], and cyclophosphamide [31]. An additional concern exists for autologous stem cell transplant in combination with high-dose chemotherapy, which is an experimental MS treatment and may decrease fertility [32].

Clinical considerations: because it is prudent to optimize a patient’s chance of conception off of DMT, strategies such as substituting barrier methods for OC, waiting for ovulation cycles to resume prior to discontinuing DMT, and referrals for evaluation of fertility if pregnancy does not result after 6 months of attempt should be considered [6].

## 4. Pregnancy

### 4.1. Pregnancy and the Risk of MS

There are multiple studies supporting the argument that pregnancy can be protective. Both a reduction in MS risk [10,11,33,34,35] and a delay in MS onset [10] has been reported. For example, Runmarker et al. found a higher risk of MS in nulliparous women compared to parous women, suggesting that pregnancy can be protective. This risk appeared to increase with older age [35]. Magyari et al. reported a 46% reduced risk of MS in Danish women during the five years following childbirth [33], while Ponsonby et al. found that a higher parity and a higher number of offspring was associated with reduced risk of a first clinical demyelinating event, and that each pregnancy conferred a decreased risk [34]. Similarly, in a case-controlled study, Rejali et al. found a significant relationship between the number of pregnancies and reduced risk of MS (OR = 0.586, 95% CI = 0.461–0.745), suggesting that higher parity also influences MS risk [11]. Holmqvist et al. found that pregnancy significantly delayed the mean age at MS onset in women who had given birth prior to disease onset. This increase in age was seen for each child born to a woman prior to MS onset, suggesting a protective effect of each pregnancy [10] which could be related to long-term epigenetic effects [36]. 

### 4.2. Pregnancy and Risk of Disease Activity

During pregnancy, the female’s immune system is reported to be more immunotolerant due to a shift in the ratio of T helper 1 and 2 cells, mediated by high levels of estrogens, especially estradiol, in addition to other important hormones including progesterone and androgens [6,10]. Some of these changes support the clinical observation of decrease relapses during pregnancy and likely also support the return of disease activity observed in months post-partum.

One of the pivotal studies to support this was reported in 1998, where Confravreux et al. prospectively reported on the natural history of MS in pregnancy. A reduction in ARR during pregnancy of 70%, especially during the third trimester, compared with the year prior to pregnancy was reported and was followed by an increase in ARR during three months post-partum. The relapse rate subsequently returned to the pre-partum relapse rate [37]. Contrary to Poser’s report [38], there was no negative effect of pregnancy on disability progression [37]. In 2004, a two-year post-partum follow up by Vukusic et al. noted that the risk of post-partum relapse was correlated to the number of relapses in the year prior to pregnancy, in the number of relapses during pregnancy, and in those with a higher Expanded Disability Status Scale (EDSS) at baseline [39]. Other studies [40,41] published worldwide have found similar trends in the pattern of relapse occurrence during pregnancy and post-partum which supports that the observations are more likely to be return of activity. 

Clinical considerations: because there is the theoretical risk for relapses to occur in the early phases of pregnancy, it is important to provide a recommendation in the clinical setting concerning treatment. This risk of early relapses has become more apparent in patients who are becoming pregnant while on fingolimod and natalizumab. 

Previous studies have reported on treatment effects with interferon beta or glatiramer acetate on relapse rates but little information is available concerning second-line therapies that are only more recently available and are not typically used during pregnancy due to safety concerns [42]. There is data to support the argument that the older treatments, such as glatiramer acetate or interferon beta had no effect [43]. There have been reports of rebound with newer and higher efficacy DMT with discontinuation of therapy prior to or during pregnancy [42]. A cross-sectional study of 99 pregnancies with nearly 90% of women treated with DMTs during the year prior to pregnancy showed a higher relapse occurrence during pregnancy than previously reported [42]. A four-fold increase in relapses during pregnancy occurred, mostly during the first and third trimesters. Relapses were primarily seen in patients treated with natalizumab and fingolimod before pregnancy. Longer DMT washout periods before pregnancy resulted in a higher probability of relapse (OR 3.9, 95% CI 1.4–10.6, *p* = 0.008), and were seen in patients previously treated with natalizumab and fingolimod who relapsed during the first trimester of pregnancy [42]. 

It is important to recognize that there are certain DMT’s that can place a woman at risk for relapse prior to or during pregnancy. Discontinuing fingolimod has been reported to increase risk of rebound disease and a suggested washout of 2 months is recommended [15]. Women taking fingolimod should be counseled regarding the risk of rebound activity, and switching to a safer medication such as glatiramer acetate or interferon beta should be considered, based on newer data from postmarking studies [42,44,45,46]. Patients taking natalizumab may also be at increased risk of rebound disease activity and switching to glatiramer acetate or interferon beta may be similarly considered [42,46]. For women on natalizumab with highly active disease, Thone et al. [45] recommends a consideration of either shortening the washout period, continuing natalizumab until conception, or continuing natalizumab during pregnancy with extended interval dosing every 6 weeks through gestation week 30, with a pediatric check upon delivery for hematologic abnormalities. This should be done on a case-by-case basis and after a discussion about the risks with each patient [45]. Continuation of natalizumab during the entire pregnancy has also been suggested in those patients with high disease activity who are at high risk of rebound [47]. However, a full discussion between patient and physician needs to occur highlighting the potential risk of fetal hematologic abnormalities reported with natalizumab use [46,47]. If exposed, we recommend that women be under the care of a high-risk obstetrician and consider delivering in a hospital where a pediatrician is accessible to assess the newborn for potential complications [47]. Other considerations for these patients include discontinuation of DMT followed by empiric treatment with prophylactic monthly high-dose corticosteroids during attempted conception after a negative pregnancy test [6], switching treatment to alemtuzumab and waiting 4 months before attempted conception [17], or switching treatment to B-cell therapy and waiting 6 months before attempted conception [18]. These studies are part of the recommendations found in the joint European Committee of Treatment and Research in Multiple Sclerosis (ECTRIMS) and European Academy of Neurology (EAN) guidelines for treatment of women with MS who wish to become pregnant [48] (Table 3). 

Relapses during pregnancy should have clinical and radiologic assessments when indicated. A brain MRI without contrast does not appear to be harmful to the fetus [49]. Treatment with prednisone, prednisolone, or methylprednisolone can be used to shorten symptom duration, as these medications are metabolized by the placenta by about 90% [50]. However, their use in the first trimester of pregnancy is considered teratogenic and an increased risk of cleft lip or palate has been reported [51].

### 4.3. Maternal and Delivery Complications

The effect of MS on pregnancy outcomes in the US and throughout the world has been previously described, with conflicting results [40,41,52,53,54,55]. A meta-analysis cited a relatively higher prevalence of abortions, cesarean sections, prematurity, and low birth weight in women with MS. However, due to the high heterogeneity between the studies, they felt the effects of regional, legal, and cultural differences could have hyper-inflated the abortion and cesarean section rates [40]. Studies using claims databases have mixed reports. A higher risk of both antenatal [53] or post-partum [52] hospitalization of mothers with MS were reported in two studies, whereas other studies found no increased risk [3,54,55]. Reports of maternal infections such as urinary tract infections (UTI) [54] and sexually transmitted diseases (STD) [3], cesarean section rates [53,54] and induction of labor [54] were increased in women with MS compared to controls. Risk of premature labor [3] and preterm delivery [55] have also been reported.

The conflicting evidence regarding pregnancy outcomes in women with MS may stem from ascertainment bias and other confounding variables. Both older age and an increased prevalence of co-morbidities in women with MS can independently affect pregnancy outcomes. There are several potential issues with using claims based databases, where coding and billing might simply reflect increased health care use in MS patients due to increased caution by their providers [41], where different databases capture differing MS subpopulations based on the health insurance included in that database, where reliance on hospital discharge summaries for identification of MS patients may not capture all patients if they are asymptomatic at the time of delivery and no code for MS is entered, and where a lack of vital background information including MS disease duration, disability, prior IV corticosteroid use during pregnancy, other medications used during pregnancy, and prior pregnancy outcomes limit interpretation of the results [3,52,53,54,55]. 

There is still much to be learned about MS and whether it is associated with adverse pregnancy outcomes. Well-designed analyses not reliant on recall, standardized and more detailed data collection, and increased participation in pregnancy registries are all vital to our future knowledge [3]. 

### 4.4. Fetal and Neonatal Complications

A majority of reported data regarding fetal and neonatal outcomes in women with MS has been reassuring [40,52,53,54,55]. Rates of malformations and neonatal deaths have been reported as low as 1.13–6.25%, which is similar to US reported rates [40,56]. Several US claims based studies have evaluated the rate of fetal malformations in newborns born to mothers with and without MS, and most [52,53,54,55], but not all [3], have not demonstrated an increased risk. Houtchens et al. alone found that a higher proportion of patients with MS than without MS had claims for acquired fetal damage (27.8% vs. 23.5% *p* = 0.002) and congenital fetal malformations (13.2% vs. 10.3% *p* = 0.004) [3]. Importantly, in this study, both MS and control groups had a higher rate of labor and delivery complications than reported in the general population [56]. Additional considerations include the fact that MS patients were older, were more likely to have other chronic medications conditions, and due to the study design, other pertinent health and pregnancy information was not available. Thus, independent predictors of adverse pregnancy outcomes were present, but were not able to be appropriately interpreted, again highlighting the need for additional information which could not be ascertained from the data [3,56].

Clinical considerations: none of the current US Food and Drug Administration (FDA) approved DMTs used to treat MS are specifically approved for women who are trying to conceive, who are currently pregnant, or who are breastfeeding. There are variations in DMT washout prior to pregnancy [15,16,17,18,19,57,58] which are summarized in Table 4 along with their associated FDA pregnancy categorization [46,59]. The recommendations differ in other countries from those in the US, especially for glatiramer acetate [48].

A summary of prescribing recommendations in pregnancy can be found in the labels of MS DMTs, including a summary of their safety, lactation warnings, and contraception guidelines. Most FDA approved MS DMTs have established pregnancy registries, and providers should encourage patients with DMT exposures during pregnancy to enroll. Interestingly, a recent retrospective study using an international MS pregnancy database found an increasing percentage of MS patients becoming pregnant on DMTs, rising from 27% in 2006 to 62% in 2016, with a median exposure duration of one month. The proportion of pre-term births or miscarriages compared to the unexposed pregnancies were no different [60]. Updated information on the safety of DMT use during pregnancy based from the FDA label, registry data and other sources may be found in the Appendix A [15,16,17,18,19,30,31,46,57,61,62,63,64,65,66,67,68,69,70,71,72,73,74,75,76,77,78]. 

## 5. Post-Partum

### 5.1. Post-Partum and the Risk of Disease Activity

Several studies have down that the post-partum period is a time of risk for increased disease activity [37,41,42]. In a cross-sectional study of 512 patients, Poser et al. found that women at six months post-partum were at increased risk of developing MS, disability progression, and of experiencing a higher number of relapses compared with during pregnancy [38]. Pregnancy may also increase the risk of conversion from radiologically isolated syndrome (RIS) to CIS [79]. In a small French cohort of 60 women with RIS, those who became pregnant developed a first neurologic event earlier or had more active MRI lesions than those who did not become pregnant [79]. Due to its small sample size and the rarity of RIS, larger studies are needed to better understand the influence of pregnancy on the risk of conversion from RIS to CIS.

As none of the current DMTs are FDA-approved for use during breastfeeding, it is important to discuss the appropriate time to resume DMT. To balance the risk of relapse to the mother with the benefit of breastfeeding to the baby in considering resuming DMT, the physician must consider a patient’s likelihood to experience post-partum disease activity. Research supports that there is a relationship between post-partum relapse rates and prior disease activity the year before pregnancy [39]. The highest predictor of post-partum relapse was a higher annualized relapse rate in the 2 years prior to pregnancy [80]. For women at increased risk of relapse, a discussion regarding benefits of resuming DMT versus the risks of relapse with breastfeeding should occur. Women who chose to forego breastfeeding to resume DMT or who simply do not wish to breastfeed should resume DMT in 7–10 days post-partum [45].

### 5.2. Breastfeeding and Disease Activity

Breastfeeding has numerous health benefits to both the mother and infant. Exclusive breastfeeding for six months should be encouraged if possible, as recommended by the World Health Organization [81]. The reported effects of breastfeeding on post-partum relapse rate in MS have varied [6,37,82,83,84] and generally the patient is advised to make an informed decision.

Several prospective studies have found a benefit with breastfeeding and in particular exclusive breastfeeding on post-partum relapse rates [37,82,83] (Table 5). Exclusive breastfeeding is thought to provoke lactational amenorrhea and ovarian suppression, which may be important in the pathophysiology behind the observed benefit in MS [85]. In a small prospective study, women who exclusively breastfed were five times less likely to experience a relapse in the year following pregnancy compared to those who did not and exclusive breastfeeding was associated with protracted lactational amenorrhea. This study was limited by its small size, but showed a consistent benefit of exclusive breastfeeding when women with more highly active disease, a source of possible confounding, were excluded from analysis [82]. A meta-analysis of the effects of breastfeeding on relapse rate reported that women who breastfed were half as likely to have a post-partum relapse than women who did not [84]. Limitations included high heterogeneity between studies, exclusivity of breastfeeding was not always defined or was simplistically defined, and recall bias in the included retrospective studies. A larger prospective study was conducted using the German MS and Pregnancy Registry that enrolled pregnant MS women based on their intention to exclusively or non-exclusively breastfeed. A lower risk of relapse in the six months post-partum was seen in women who chose to breastfeed exclusively compared to those who did not, and compared to those who resumed DMT within 30 days [83]. 

A recent study examined the effects of breastfeeding on the risk of MS in the nursing mother. The authors hypothesized that breast-feeding leads to anovulation, which confers decreased MS risk. Interestingly, in this matched case-control study, a cumulative duration of breastfeeding for ≥15 months was associated with a reduced risk of developing MS (adjusted OR 0.47, 95% CI 0.28–0.77; *p* = 0.003) compared to <4 months of breastfeeding, concluding that longer breastfeeding confers protection of developing MS to the nursing mother. Interestingly, the authors did not find an association between ovulatory years and MS risk. Limitations of this study included recall bias, the lack of hypothesized association of ovulatory years with MS risk, and lack of information as to why women chose not to breastfeed [86]. The effect of breastfeeding on MS risk requires further evaluation.

Clinical considerations: when resuming DMT, it is important to consider that the patients with more highly active disease pre-pregnancy and/or during pregnancy are more likely to chose not to breastfeed and to resume DMT post-partum and thus confound the above results [6], although the authors did account for this as best they could [82,83]. A larger, prospective study evaluating the effects of exclusive breastfeeding on post-partum relapse is needed. At the least, breastfeeding appears to be safe in women with MS, and exclusive breastfeeding may be beneficial until food supplementation is introduced into the infant’s diet. For women who wish to breastfeed, the data support greater benefit from exclusive and without supplementation. 

In patients who wish to breastfeed but are at a high risk of post-partum relapse, a protective strategy is using monthly corticosteroids [87]. Although only about 10% of an infant’s endogenous corticosteroid production is ingested from breast milk, the long-term risks are unknown. Thus, a four-hour period of delayed breastfeeding, or “pumping and dumping,” is recommended [88]. For monitoring purposes, routine screening MRI with gadolinium after delivery, again followed by delayed breastfeeding for 24 h, in breastfeeding mothers is recommended to help identify subclinical MS activity, and those who may need DMT earlier than anticipated [6]. 

### 5.3. Newborn Care

Women with MS are faced with balancing their own needs with those of their newborn child during the post-partum period. Sleep management, rehabilitation (often in the form of pelvic floor exercises), and resumption of DMT are important maternal considerations. Family or paid assistance should be encouraged. Women should also be screened for post-partum depression due to its high overall prevalence in the population, with the strongest risk factor of having baseline depression [5].

## 6. Conclusions

Women with MS are becoming pregnant with increasing rates, while newer and more effective DMTs are used to treat MS patients. Knowledge about safe and effective DMT use and the pattern of disease activity around pregnancy is vital. Decisions regarding breastfeeding and DMT resumption post-partum are shared between patient and physician, but with appropriate discussion about risks and benefits. Special consideration should be taken with women who have highly active disease, or who take medications that might cause rebound disease activity. Generally speaking, women should feel reassured and confident about the management of their MS in the reproductive years. 

## Figures and Tables

**Figure 1 biomedicines-07-00032-f001:**
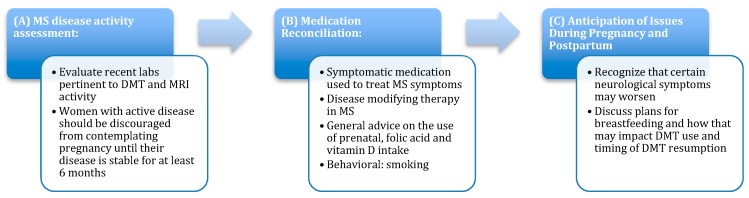
Pre-pregnancy care. (**A**) Multiple sclerosis (MS) disease activity assessment, (**B**) medication reconciliation, (**C**) anticipation of issues during pregnancy and post-partum.

**Table 1 biomedicines-07-00032-t001:** Summary of prescribing contraception recommendations for MS disease-modifying treatments (DMTs).

Disease Modifying Treatment (DMT)	Prescribing Contraception Recommendations *
Interferon beta	N/A
Glatiramer acetate	N/A
Fingolimod	Effective contraception during treatment and two months following therapy [15]
Dimethyl fumarate	N/A
Teriflunomide	Effective contraception during treatment and until plasma concentrations of teriflunomide are less than 0.02 mg/L [16]
Natalizumab	N/A
Alemtuzumab	Effective contraceptive measures during treatment and for 4 months following that course of treatment [17]
Ocrelizumab	Effective contraception during therapy and for 6 months after the last infusion [18]
Mitoxantrone	Women should not become pregnant during therapy [19]
Cyclophosphamide **	Effective contraception during therapy and for up to 1 year after completion of treatment [20]

* Contraception recommendations included if provided in the prescribing label. ** Not Food and Drug Administration (FDA) approved for use in MS.

**Table 2 biomedicines-07-00032-t002:** Summary of reported MS DMT effects on fertility.

DMT	Fertility
Interferon beta	Reduced fertility in animals; no information in humans [30]
Glatiramer acetate	No effects
Fingolimod	No effects
Dimethyl Fumarate	No effects
Teriflunomide	No effects
Natalizumab	Reduced fertility in animals; no information in humans [31]
Alemtuzumab	Reduced fertility in animals; no information in humans [17]
Ocrelizumab	No effects
Mitoxantrone	Amenorrhea and transient azoospermia have been reported [31]
Cyclophosphamide *	Amenorrhea and transient azoospermia have been reported [31]

* Not FDA approved for the treatment of MS.

**Table 3 biomedicines-07-00032-t003:** European Committee of Treatment and Research in Multiple Sclerosis (ECTRIMS)/European Academy of Neurology (EAN) Guidelines for management of DMT in women with MS in pregnancy.

Only Glatiramer Acetate 20 mg/mL is Approved for Use during Pregnancy (Consensus) [48].
For women who are at risk of disease reactivation and planning pregnancy: ○ Consider switching the glatiramer acetate or interferon beta until conception is confirmed (weak) [48]. ○ For specific cases in women with active disease, consider continuing treatment throughout pregnancy (weak) [48].
Delaying pregnancy is advised for women with persistent, highly active disease. ○ Women who become pregnant unintentionally or despite recommendation, treatment with natalizumab during pregnancy can be considered after counseling about the potential consequences (weak) [48]. ○ For planned pregnancies with very active disease, alemtuzumab can be considered, but treatment must be followed by 4 months of effective contraception (weak) [48].

**Table 4 biomedicines-07-00032-t004:** FDA pregnancy categories and recommended washout periods for MS DMTs.

DMT	FDA Pregnancy Category	Recommended Washout Period
Interferon beta	Category C [46]	0–1 Menstrual cycles [57,58]
Glatiramer acetate	Category B [46]	0–1 Menstrual cycles [57,58]
Fingolimod	Category C [46]	2 Menstrual cycles [15]
Dimethyl Fumarate	Category C [46]	0–1 Menstrual cycles [57,58]
Teriflunomide	Category X [46]	Either: (1) Wait 24 Menstrual cycles, OR (2) Perform accelerated elimination until plasma concentration <0.02 mg/dL [16]
Natalizumab *	Category C [46]	1–3 Menstrual cycles [58]
Alemtuzumab	Category C [46]	4 Menstrual cycles [17]
Ocrelizumab	N/A	6 Menstrual cycles [18]
Mitoxantrone	Category D [46]	6 Menstrual cycles [19]

* Natalizumab: Case by case decision should be made in females with highly active disease or with history of severe natalizumab withdrawal relapse after full discussion of the risks and benefits with the patient.

**Table 5 biomedicines-07-00032-t005:** Annualized relapse rates (ARR) in breastfeeding.

Study	Breastfeeding	ARR Pre-Pregnancy	ARR During Pregnancy	ARR Post Partum
Confavreux et al. [37]	Yes	0.6	0.3	1–3 months: 1.2 4–6 months: 0.9
	No	0.8	0.5	1–3 months: 1.3 4–6 months: 1.0
Hellwig et al. [83]	Exclusive	N/A	0.22	0.48
	Non-Exclusive	N/A	0.36	0.77
Langer-Gould et al. [82]	Exclusive	0.57	0.19	0.36
	Non-Exclusive	0.83	0.18	0.87

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
