# Peer review of "Women’s Health: Contemporary Management of MS in Pregnancy and Post-Partum"

_biomedicines, 2019, doi:10.3390/biomedicines7020032_

Reviewer 1 Report

This is a well structured and comprehensive review of the issues surrounding pregnancy and MS. My comments are mainly concerned with clarifying the language.

line 22- "a low-risk time for MS activity" - I suggest change the whole sentence to read "There is now considerable evidence that activity of MS is reduced in pregnancy."

Line 30. I think that the comment that the age of onset corresponds to the child-bearing age needs to be a separate sentence. 

Line 30 "Nulliparity, multiparty and the stages of
31 pregnancy and lactation support that these actions contribute to some aspect of the disease." - this sentence is unclear- suggest delete

Line 43: consider adding this reference which shows that pregnancy has long-term epigenetic effects on MS genes   https://doi.org/10.1016/j.jneuroim.2018.12.004

Line 46 - I don't think that the "act of pregnancy"  is a good term-  maybe change to :during pregnancy" 

Line 93- give the reference to this study

Line 101- I thin that this needs to be a stronger statement- rather than say there are options- say that there are some important considerations

Line 114- this sentence uses consideration and consider- suggest re-word

Line 132 - instead of "proper" maybe use "appropriate"

Line 250  - please consider adding the recent study  https://doi.org/10.1016/j.msard.2019.01.003

Line 287: please consider acknowledging that the recommendations in other countries differ form those in the US- esp for glatiramer

Line 324: instead of one- could you say " the physician must consider"

Line 327- can you change whereas to "however"

Line 330 - change should be held to should occur

Line 392: change "one's disease provider"  - here one is the patient. Change to should be evaluated. 

Line 407- change to "potential for MS symptoms to worsen"

Author Response

Please find attached word document.

Reviewer 2 Report

The topic is interesting. The objective is not clear, it should be formulated more clearly. The justification for the study is weak, perhaps providing some prevalence data to help visualize the magnitude of the problem.

It lacks a clear methodology. There is no methodology (consulted databases, Mesh terms, etc.). Nor is there a discussion . There are no limitations, no strengths, etc.

The idea is good, the content also but the structure of the article should improve.

Author Response

Please find attached word document.

Round  2

Reviewer 2 Report

The authors have improved the article. Now it's clearer